# Boosting the Local Invariance for Better Adversarial Transferability

## Abstract

Transfer-based attacks pose a significant threat to real-world applications by directly targeting victim models with adversarial examples generated on surrogate models. While numerous approaches have been proposed to enhance adversarial transferability, existing works often overlook the intrinsic relationship between adversarial perturbations and input images. In this work, we find that adversarial perturbation often exhibits poor translation invariance for a given clean image and model, which is attributed to local invariance. Through empirical analysis, we demonstrate that there is a positive correlation between the local invariance of adversarial perturbations w.r.t the input image and their transferability across different models. Based on this finding, we propose a general adversarial transferability boosting technique called Local Invariance Boosting approach (LI-Boost). Extensive experiments on the standard ImageNet dataset demonstrate that LI-Boost could significantly boost various types of transfer-based attacks (e.g., gradient-based, input transformation-based, model-related, advanced objective function, ensemble, etc.) on CNNs, ViTs, and defense mechanisms. Our approach presents a promising direction for future research in improving adversarial transferability across different models.

## 1 Introduction

Deep Neural Networks (DNNs)( He et al. (2016); Krizhevsky et al. (2012); Vaswani et al. (2017)) have achieved substantial success across various tasks, e.g., image recognition( Szegedy et al. (2016); Huang et al. (2017); Dosovitskiy et al. (2021)), image generation( Rombach et al. (2022); Ramesh et al. (2022)), and large language model( Brown et al. (2020); Touvron et al. (2023)), etc. However, researchers have identified that DNNs are vulnerable to adversarial examples( Szegedy et al. (2014); Goodfellow et al. (2015)), where small, often imperceptible perturbations can deceive the model into making incorrect predictions. This vulnerability poses a serious risk to real-world DNN-based applications, particularly in security-sensitive domains such as face verification( Sharif et al. (2016)) and autonomous driving( Eykholt et al. (2018)). Consequently, adversarial attack( Goodfellow et al. (2015); Moosavi-Dezfooli et al. (2016); Kurakin et al. (2017); Wang et al. (2019)) and defense( Madry et al. (2018); Shafahi et al. (2019); Cohen et al. (2019); Naseer et al. (2020)) strategies have attracted extensive research interest. One of the intriguing characteristics of adversarial examples is their transferability across different models, where adversarial examples generated on a surrogate model can deceive previously unseen victim models( Liu et al. (2017); Dong et al. (2018)). Unlike other attack strategies, transfer-based attacks do not necessitate access to the information of victim models, making them a particularly practical and serious threat to real-world DNN applications. Given these potential risks, extensive research has been conducted to enhance the transferability of adversarial attacks( Wu et al. (2020); Wang et al. (2021c); Lin et al. (2020); Wang & He (2021); Xie et al. (2019)).

Existing transfer-based attacks can be broadly categorized into five types: 1) **Gradient-based attacks**( Dong et al. (2018); Lin et al. (2020); Wang & He (2021)), which typically incorporate various momentum techniques to stabilize the optimization process and improve convergence. 2) **Input transformation-based attacks**( Xie et al. (2019); Wang et al. (2021a; 2024a)), which apply transformations to the input image to enhance the diversity of gradients for more effective optimization. 3) **Model-related attacks**( Wu et al. (2020); Guo et al. (2020); Wang et al. (2023a)), which introduce model-specific modifications during the forward or backward propagation stages. 4) **Advanced ob-**

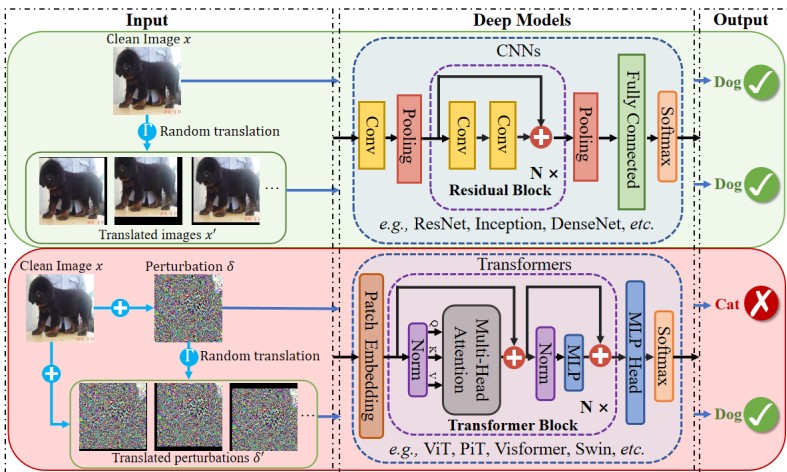

Figure 1: The impact of translation invariance of clean image and adversarial perturbation. The translated clean image can be correctly recognized by deep models (either CNNs or ViTs) while the translated adversarial perturbations cannot consistently fool the deep models.

**jective functions**( Wang et al. (2021c); Huang et al. (2019); Li et al. (2023)), which design novel objective functions using the mid-layer features. 5) **Ensemble attacks**( Liu et al. (2017); Dong et al. (2018); Xiong et al. (2022)), which target multiple models simultaneously to increase adversarial transferability. Notably, these approaches directly optimize the perturbation w.r.t the input image, without accounting for the inherent relationship between the perturbation and the input itself.

It is widely known that clean images are consistently and accurately classified by various deep learning models and exhibit robust translation invariance on the same model. As shown in Fig. 1, however, we find that adversarial perturbations exhibit significantly weaker translation invariance for the same clean image and model. This observation is counterintuitive, given the inherent similarity among the local regions of clean images. We hypothesize and empirically validate that the local invariance of adversarial perturbation w.r.t clean image for a given model is positively correlated to its adversarial transferability across different models. Building on this insight, we introduce a novel and generalizable framework to enhance the transferability of various transfer-based attacks. Our contributions are summarized as follows:

- We introduce local invariance for adversarial perturbations and unveil the underlying relationship between local invariance on the surrogate model and adversarial transferability across different models, which provides new insights to boost adversarial transferability across various models.

- We propose a novel and general boosting approach called LI-Boost to enhance adversarial transferability. Specifically, at each iteration, LI-Boost optimizes the adversarial perturbation using the gradient of adversarial examples with several translated perturbations to enhance the local invariance.

- Extensive experiments on the ImageNet dataset demonstrate that LI-Boost can effectively boost various types of transfer-based attacks on either CNNs or ViTs w/wo defense mechanisms, showing its generality and superiority in various scenarios.

## 2 RELATED WORK

In this section, we provide a brief overview of existing adversarial attack and defense approaches.

### 2.1 ADVERSARIAL ATTACKS

After identifying the vulnerability of DNNs against adversarial examples( Szegedy et al. (2014)), numerous adversarial attacks have emerged( Moosavi-Dezfooli et al. (2016); Madry et al. (2018); Wang et al. (2019)). White-box attacks( Goodfellow et al. (2015); Kurakin et al. (2017); Croce & Hein (2020)), which have full access to the target model (e.g., gradients, architectures, and logits, etc.), are widely used to assess the robustness of DNNs. In contrast, black-box attacks, which have limited access to the target model, pose more severe threats to real-world DNN-deployed applications. Black-box attacks can be further categorized into score-based attacks( Uesato et al. (2018);

Guo et al. (2019); Andriushchenko et al. (2020)), decision-based attacks( Li et al. (2020); Wang et al. (2022); Maho et al. (2021)) and transfer-based attacks( Liu et al. (2017); Wang & He (2021)). Among these, transfer-based attacks, where the adversarial examples generated on surrogate models are used to attack the target model without any direct access, have garnered significant research interest( Xie et al. (2019); Gao et al. (2020); Zhang et al. (2023a); Wang et al. (2023c); Zhang et al. (2024b); Naseer et al. (2022); Li et al. (2023); Zhang et al. (2022a; 2023b; 2024a)).

**Gradient-based attacks** (e.g., FGSM Goodfellow et al. (2015), I-FGSM Kurakin et al. (2017)) are popular white-box attacks that exhibits superior white-box attack performance but poor transferability. To boost adversarial transferability, various approaches integrate momentum to stabilize the optimization( Lin et al. (2020); Qin et al. (2022); Wang et al. (2021b)). For instance, MI-FGSM( Dong et al. (2018)) first integrates the momentum into I-FGSM and achieves much higher transferability. VMI-FGSM( Wang & He (2021)) further refines gradient variance to stabilize the update direction. PGN( Ge et al. (2023)) introduces a penalized gradient norm to the original loss function, producing adversarial examples in flatter local regions with improved transferability across models. MUMODIG( Ren et al. (2025b)) improves transferability through generating integration paths using diverse baseline samples and enforcing the monotonicity of each path.

Numerous **input transformation-based attacks** have been proposed to boost adversarial transferability( Zou et al. (2020); Dong et al. (2019); Wang et al. (2024a)). DIM( Xie et al. (2019)) improves transferability by randomly resizing and padding the input image before the gradient calculation. *Admix*( Wang et al. (2021a)) enhances diversity by combining the original image with a second image from a distinct category to generate more diverse perturbations. SIA( Wang et al. (2023b)) applies various transformations to the blocks of the input image to increase diversity while maintaining its structural integrity. BSR( Wang et al. (2024a)) splits the image into blocks then shuffles and randomly rotates them to enhance transferability.

Additionally, **model-related attacks** often modify the architecture of surrogate model for enhanced transferability. For example, Linbp ( Guo et al. (2020)) modifies the backward propagation process by setting the gradient of the ReLU activation function to a constant value of 1 and scaling the gradients of residual blocks. SGM ( Wu et al. (2020)) prioritizes the gradients from skip connections over those from residual modules to improve transferability. BPA ( Wang et al. (2023a)) introduces a non-monotonic function as the derivative of ReLU and integrates a temperature-controlled softmax function to activate the truncated gradient for better transferability. VDC ( Zhang et al. (2024a)) imports virtual dense connections for dense gradient back-propagation in attention maps and MLP blocks based on the forward propagation for vision transformers. FPR ( Ren et al. (2025a)) refines the forward propagation through diversifing the attention map and accumulating the output token embedding using momentum mechanism.

**Advanced objective functions** often perturb mid-layer features to improve transferability( Wang et al. (2023c); Zhang et al. (2022a;b)). For instance, ILA ( Huang et al. (2019)) enhances the similarity of feature differences between an adversarial example and its benign counterpart on a pre-specified layer of the source model. FIA ( Wang et al. (2021c)) disrupts object-aware features that significantly influence model decisions to calculate the aggregated gradients for updating the perturbation. ILPD ( Li et al. (2023)) amplifies the magnitude of perturbations in the adversarial direction within intermediate layers by incorporating perturbation decay in a single-stage optimization framework. BFA ( Wang et al. (2024b)) employs fitted gradients and feature maps to destroy the black-box features.

**Ensemble attacks** simultaneously generate adversarial examples on multiple surrogate models to enhance adversarial transferability. Dong et al. (2018) aggregate the logits from all surrogate models to generate adversarial examples. SVRE ( Xiong et al. (2022)) adopts the stochastic variance to reduce gradient variance between various models and escape poor local optima during the update process. CWA ( Chen et al. (2024)) identifies shared vulnerabilities across an ensemble of models to improve transferability.

### 2.2 ADVERSARIAL DEFENSE

Numerous defenses have been proposed to mitigate the threat of adversarial examples. Adversarial training ( Goodfellow et al. (2015); Tramèr et al. (2018); Madry et al. (2018); Shafahi et al. (2019)) adopts the adversarial examples during the training process, which has been one of the most effective methods to improve the model's robustness. Fast-AT ( Wong et al. (2020)) adopts a single

iteration to generate adversarial examples for training, which can significantly boost adversarial robustness. Guo et al. (2018) employed various image transformations (e.g., JPEG compression, etc.) to preprocess inputs before feeding them into the models. Liao et al. (2018) propose the high-level representation guided denoiser (HGD) by minimizing the difference between the model's outputs on clean and denoised images. Naseer et al. (2020) developed a Neural Representation Purifier (NRP) trained using a self-supervised adversarial training method to purify input images. Several certified defense methods offer verifiable defense capabilities, such as randomized smoothing (RS) ( Cohen et al. (2019)). Besides, diffusion models for purification (DiffPure) ( Nie et al. (2022)) exhibit an excellent potential for adversarial defense.

## 3 METHODOLOGY

### 3.1 PRELIMINARIES

Given a victim model $f$ with parameters $\theta$ and a clean image $x \in \mathcal{X}$ with ground-truth label $y$, where $x$ is in $d$ dimensions and $\mathcal{X}$ denotes all the legitimate images, adversarial attacks seek to identify an adversarial example $x + \delta \in \mathcal{X}$ such that:

$$f(x; \theta) \neq f(x + \delta; \theta) \quad \text{s.t.} \quad \|\delta\|_p \leq \epsilon. \tag{1}$$

Here $\epsilon$ represents the perturbation budget, $\delta$ is the perturbation of $x$, and $\|\cdot\|_p$ is the $\ell_p$-norm distance. In this work, we adopt $\ell_\infty$ distance to align with existing works. To generate such a perturbation, the adversary typically maximizes the loss function $J$ (e.g., cross-entropy loss) of the target model, which can be formalized as:

$$\delta = \arg \max_{\|\delta\|_p \leq \epsilon} J(x + \delta, y; \theta). \tag{2}$$

The transferability of adversarial examples generated on the surrogate model when applied to the victim model $f$ can be evaluated by the attack success rates (ASR) as follows:

$$ASR = \frac{1}{|\mathcal{X}|} \sum_{x \in \mathcal{X}} \mathbb{I}[f(x) \neq f(x + \delta)], \tag{3}$$

where $\delta$ is generated on surrogate model $f_s$ w.r.t the input image $x$ and $\mathbb{I}(\cdot)$ is the indicator function.

### 3.2 MOTIVATION

Deep neural networks (DNNs) with different architectures often exhibit the ability to consistently recognize the same image, demonstrating the model-independent semantic consistency of clean images. In addition, DNNs are known for their strong translation invariance ( Jaderberg et al. (2015); Kauderer-Abrams (2018)), wherein they reliably produce accurate predictions across translated versions of an image. This behavior mirrors the human visual system that the translated images can be correctly recognized, as translation does not fundamentally alter the images' semantic content.

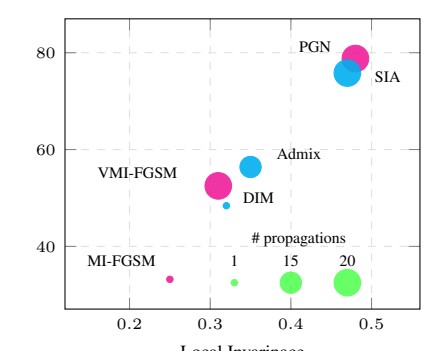

Figure 2: The local invariance ($k = 5$) on RN-50 and average attack success rates (%) on nine models of various transfer-based attacks.

Adversarial transferability refers to the ability of adversarial examples generated on the surrogate model to successfully deceive other models. This concept parallels the observation that clean images are often classified correctly by various models. However, existing adversarial examples often exhibit weak transferability across different models, particularly between CNNs and ViTs. Besides, as shown in Fig. 1, we observe that adversarial perturbations also exhibit poor translation invariance for a given clean image and DNN. This observation contradicts human intuition, which suggests that local regions of an image should retain consistent semantic features. In contrast, the corresponding adversarial perturbations vary significantly. For example, while the pixels of a dog's ear are visually similar, the associated perturbations vary substantially. This indicates that the perturbations not only overfit the victim model but also become highly sensitive to pixel positions within the image.

This finding inspires us that translation invariance may be beneficial for enhancing adversarial transferability. To validate this assumption, we first define the local invariance of adversarial perturbation $\delta$ to quantify translation invariance as follows:

**Definition 1** (Local Invariance). *Given an adversarial perturbation $\delta$ for the input image $x \in \mathcal{X}$ and surrogate model $f_s$, the local invariance of perturbation is quantified as:*

$$\mathcal{I}(x, \delta, k) = \frac{\sum_{-k \leq i,j \leq k} \mathbb{I}\left[f_s(x) \neq f_s(x + \Gamma(\delta, i, j))\right]}{(2k+1)^2},$$

*where $\Gamma(\delta, i, j)$ denotes the translation operator that translates $\delta$ by $i$ pixels horizontally and $j$ pixels vertically, and $k$ represents the upper bound of translated pixels.*

We have calculated the average local invariance of adversarial perturbations generated by various transfer-based attacks. As shown in Fig. 2, we observe that improved adversarial transferability is often associated with better local invariance. Based on this observation, we conclude that the local invariance of adversarial perturbations serves as an indicator of their transferability across different models. Furthermore, enhancing local invariance appears to positively influence the transferability of these adversarial perturbations.

### 3.3 LOCAL INVARIANCE BOOSTING APPROACH

Building on the above analysis, we propose a novel attack approach called Local Invariance Boosting approach (LI-Boost), which enhances the local invariance of adversarial perturbations w.r.t the clean image to improve transferability across various models. Specifically, we can formulate the problem as follows:

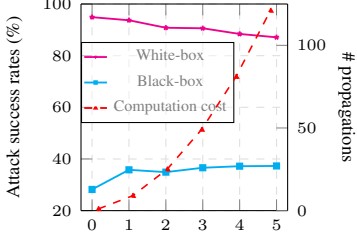

Figure 3: Attack success rates (%) and number of forward and backward propagations of MI-FGSM with Eq. equation 4 using various $k$.

$$\delta = \arg \max_{\|\delta\|_p \leq \epsilon} \left[ \min_{\delta' \in \{\Gamma(\delta, i, j) | -k \leq i, j \leq k\}} J(x + \delta', y; \theta) \right], \quad (4)$$

where $k$ represents the maximum number of pixels by which the perturbation is translated. To assess the effectiveness of this approach, we employ MI-FGSM to solve Eq. equation 4 using various $k$. We choose ResNet-50 as the white-box setting and other 8 models illustrated in Sec. 4.1 as the black-box settings. As shown in Fig. 2, adversarial transferability consistently improves when enhancing the local invariance. However, it is important to note that this enhancement comes at a cost: the performance of white-box attacks deteriorates since generating such perturbation is hard. Also, the computational complexity escalates significantly. Specifically, the number of forward and backward propagations required for each update scales quadratically with $k$, resulting in a significant increase in computational time as $k$ grows. For instance, $3 \times 3 = 9$ forward and backward propagations are needed for $k = 1$ whereas $5 \times 5 = 25$ are required for $k = 2$. This results in progressively less efficient attack computations as $k$ increases. To enhance computational efficiency without sacrificing the attack effectiveness, we randomly sample multiple translated perturbations for each update. Specifically, the gradient is computed as follows:

$$\bar{g} = \frac{1}{N} \sum_{n=1}^{N} \nabla_\delta J(x + \Gamma(\delta, i, j), y; \theta), \quad (5)$$

where $i, j$ are randomly sampled from $[-k, k]$ with $k$ is a predefined parameter of the upper bound of translated pixels, and $N$ denotes the total number of sampled perturbations. The selection of an appropriate $N$ is crucial for balancing the trade-off between attack efficiency and effectiveness. It is important to note that LI-Boost is a general boosting technique applicable to a variety of attacks. As an example, we incorporate LI-Boost into the MI-FGSM, denoted as LI-Boost-MI. The details are summarized in Algorithm 1.

## 4 EXPERIMENTS

Here we conduct extensive evaluations on ImageNet dataset to validate the effectiveness of LI-Boost. We first specify our experiment setup, then we conduct a series of experiments on five categories of transfer-based attacks. Finally, we provide ablation studies to investigate the behavior of LI-Boost.

### 4.1 EXPERIMENTAL SETUP

**Dataset.** We evaluate the proposed LI-Boost using 5,000 images from the validation set of the ImageNet dataset( Russakovsky et al. (2015)), covering 1,000 categories.

---

**Algorithm 1** LI-Boost-MI

---

**Input**: Victim model $f$ with the loss function $J$; a raw image $x$ with ground-truth label $y$; perturbation budget $\epsilon$; decay factor $\mu$; number of iterations $T$; number of sampled perturbations $N$; upper bound of translated pixels $k$

**Parameter**: $\alpha = \epsilon/T, g_0 = 0, \delta_0 = 0$

**Output**: Perturbation $\delta$

1: **for** $t = 1$ to $T$ **do**
2:     Calculate the gradient $\bar{g}_t$ w.r.t $\delta_t$ using Eq. 5
3:     Update the momentum:
    $g_t = \mu \cdot g_{t-1} + \frac{\bar{g}_t}{||\bar{g}_t||_\infty}$
4:     Update the adversarial perturbation:
    $\delta_t = \mathrm{clamp}(\delta_{t-1} + \alpha \cdot \mathrm{sign}(g_t), -\epsilon, \epsilon)$
5: **end for**
6: **return** $\delta = \delta_T$

---

Table 1: Attack success rates (%) on nine models and five defense approaches of various **gradient-based attacks** w/wo LI-Boost. The adversarial examples are crafted on ResNet-50. * indicates the white-box model.

| Gradient-based Attacks | CNNs | | | | | ViTs | | | | Defenses | | | | |
|---|---|---|---|---|---|---|---|---|---|---|---|---|---|---|
| | RN-50 | Inc-v3 | MN-v3 | DN-121 | FSNet | ViT | PiT | Visformer | Swin | AT | HGD | RS | NRP | DiffPure |
| MI-FGSM | 94.9* | 34.5 | 40.6 | 45.9 | 25.2 | 10.5 | 18.0 | 23.1 | 27.8 | 33.7 | 19.2 | 21.9 | 25.1 | 13.8 |
| LI-Boost-MI | 97.0* | 45.3 | 55.2 | 62.0 | 41.8 | 19.1 | 29.1 | 38.7 | 41.9 | 34.3 | 33.3 | 23.8 | 32.2 | 21.4 |
| VMI-FGSM | 97.5* | 54.4 | 58.0 | 66.0 | 51.2 | 28.6 | 40.5 | 46.5 | 49.4 | 36.0 | 44.4 | 25.3 | 44.1 | 21.9 |
| LI-Boost-VMI | 99.3* | 67.0 | 71.4 | 79.3 | 65.9 | 39.7 | 53.7 | 61.4 | 62.6 | 37.9 | 59.1 | 30.4 | 60.4 | 36.8 |
| PGN | 99.1* | 84.2 | 86.4 | 91.6 | 81.5 | 54.8 | 69.7 | 77.0 | 78.8 | 46.2 | 78.3 | 41.4 | 79.7 | 48.6 |
| LI-Boost-PGN | 98.7* | 86.4 | 87.8 | 92.1 | 83.9 | 62.1 | 74.3 | 80.2 | 81.3 | 50.4 | 82.3 | 50.8 | 84.8 | 65.4 |
| MUMODIG | 97.1* | 72.8 | 78.4 | 84.7 | 72.1 | 42.9 | 58.6 | 67.7 | 66.4 | 37.6 | 68.2 | 26.4 | 52.2 | 26.7 |
| LI-Boost-MUMODIG | 98.6* | 83.4 | 85.6 | 90.9 | 81.8 | 60.0 | 72.8 | 80.0 | 77.7 | 41.3 | 78.7 | 33.4 | 90.2 | 45.8 |

**Models.** To validate its effectiveness, we adopt various architectures as the victim models, including five CNNs, i.e., ResNet-50 ((RN-50) He et al. (2016)), Inception-v3 ((Inc-v3) Szegedy et al. (2016)), MobileNet-v3((MN-v3) Howard et al. (2019)), DenseNet-121 ((DN-121) Huang et al. (2017)), FasterNet ((FSNet) Chen et al. (2023)) and four ViTs, i.e., ViT( Dosovitskiy et al. (2021)), PiT( Heo et al. (2021)), Visformer( Chen et al. (2021)), Swin( Liu et al. (2021)). To further substantiate the effectiveness of LI-Boost, we also consider five state-of-the-art defense mechanisms, namely AT( Wong et al. (2020)), HGD( Liao et al. (2018)), RS( Cohen et al. (2019)), NRP( Naseer et al. (2020)) and DiffPure( Nie et al. (2022)).

**Baselines.** To comprehensively assess the generality of LI-Boost, we establish several baselines encompassing multiple categories of transfer-based attacks, including **gradient-based attacks** (MI-FGSM Dong et al. (2018), VMI-FGSM Wang & He (2021), PGN Ge et al. (2023)),MUMODIG Ren et al. (2025b), **input transformation-based attacks** (DIM Xie et al. (2019), *Admix* Wang et al. (2021a), SIA Wang et al. (2023b), BSR Wang et al. (2024a)), **model-related attacks** (SGM Wu et al. (2020), Linbp Guo et al. (2020), BPA Wang et al. (2023a), VDC Zhang et al. (2024a)), FPR Ren et al. (2025a), **advanced objective functions** (ILA Huang et al. (2019), FIA Wang et al. (2021c), ILPD Li et al. (2023), BFA Wang et al. (2024b)) and **ensemble attack** Dong et al. (2018). For consistency, MI-FGSM is adopted as the default backbone baseline across all experiments.

**Evaluation.** We employ the attack success rates to assess the efficacy of adversarial attacks. To ensure a fair and consistent comparison across different attacks, we adopt a common attack setting with the perturbation budget $\epsilon = 16/255$, number of iterations $T = 10$, step size $\alpha = \epsilon/T$ and the decay factor $\mu = 1.0$. We adopt $k = 6$, $N = 30$, and Logarithmic distribution to sample the translated perturbations for LI-Boost. All the baselines adopt the default parameters as in their original papers, which are detailed in the Appendix Material A.6 and all experiments are conducted on a single RTX4090 GPU with 24 GB of VRAM.

### 4.2 EVALUATION ON GRADIENT-BASED ATTACKS

To validate the effectiveness of our proposed LI-Boost, we first integrate it into various gradient-based attacks, i.e., MI-FGSM, VMI-FGSM, PGN and MUMODIG. We generate the adversarial examples on ResNet-50 and evaluate the transferability on the other CNNs, ViTs and defense methods. The results are summarized in Tab. 1, and the results on other models are summarized in the Appendix Tab. 8. As we can observe, LI-Boost significantly improves the white-box attack per-

Table 2: Attack success rates (%) on nine models and five defense approaches of various **input transformation-based attacks** w/wo LI-Boost. The adversarial examples are crafted on ResNet-50. * indicates the white-box model.

| Input Transforma-tion-based Attacks | CNNs | | | | | ViTs | | | | Defenses | | | | |
|---|---|---|---|---|---|---|---|---|---|---|---|---|---|---|
| | RN-50 | Inc-v3 | MN-v3 | DN-121 | FSNet | ViT | PiT | Visformer | Swin | AT | HGD | RS | NRP | DiffPure |
| DIM | 92.7* | 52.4 | 56.7 | 64.4 | 46.9 | 23.9 | 35.0 | 42.1 | 43.6 | 34.9 | 40.3 | 23.6 | 33.6 | 19.2 |
| LI-Boost-DIM | 98.1* | 61.0 | 68.3 | 75.8 | 61.1 | 36.0 | 47.7 | 57.2 | 56.4 | 36.3 | 54.9 | 26.9 | 42.2 | 30.4 |
| *Admix* | 99.3* | 59.4 | 67.4 | 77.6 | 54.6 | 27.7 | 41.8 | 52.5 | 53.3 | 35.7 | 47.9 | 24.6 | 44.4 | 20.8 |
| LI-Boost-*Admix* | 99.5* | 71.7 | 80.5 | 86.5 | 73.9 | 44.8 | 58.5 | 70.5 | 69.2 | 38.4 | 66.9 | 30.9 | 57.7 | 38.4 |
| SIA | 99.3* | 76.2 | 89.1 | 92.9 | 81.3 | 43.5 | 66.8 | 78.4 | 76.6 | 38.0 | 71.0 | 27.2 | 57.0 | 25.4 |
| LI-Boost-SIA | 99.8* | 87.0 | 95.1 | 96.8 | 91.8 | 64.0 | 81.2 | 90.3 | 88.1 | 42.2 | 86.8 | 36.0 | 71.2 | 45.4 |
| BSR | 98.6* | 84.6 | 92.8 | 95.7 | 87.5 | 53.1 | 75.5 | 84.7 | 81.4 | 39.2 | 81.4 | 28.7 | 58.2 | 31.3 |
| LI-Boost-BSR | 99.2* | 91.3 | 96.4 | 97.8 | 94.5 | 70.6 | 85.1 | 93.6 | 90.6 | 43.2 | 91.6 | 38.1 | 71.7 | 51.0 |

Table 3: Attack success rates (%) on nine models and five defense approaches of various **model-related attacks** w/wo LI-Boost. The adversarial examples are crafted on ResNet-50, except for VDC and FPR, which are based on ViT. * indicates the white-box model.

| Model-related Attacks | CNNs | | | | | ViTs | | | | Defenses | | | | |
|---|---|---|---|---|---|---|---|---|---|---|---|---|---|---|
| | RN-50 | Inc-v3 | MN-v3 | DN-121 | FSNet | ViT | PiT | Visformer | Swin | AT | HGD | RS | NRP | DiffPure |
| SGM | 99.5* | 44.8 | 57.2 | 61.3 | 31.8 | 15.0 | 27.7 | 33.4 | 38.8 | 35.0 | 22.8 | 23.3 | 29.3 | 14.2 |
| LI-Boost-SGM | 100.0* | 61.5 | 78.4 | 82.0 | 46.7 | 29.1 | 46.4 | 57.6 | 61.0 | 36.8 | 48.0 | 27.1 | 41.3 | 25.4 |
| Linbp | 89.2* | 44.4 | 55.8 | 62.3 | 29.0 | 9.6 | 17.2 | 28.6 | 31.8 | 34.5 | 24.2 | 22.7 | 27.7 | 22.0 |
| LI-Boost-Linbp | 99.2* | 60.1 | 76.9 | 85.4 | 52.4 | 15.2 | 25.5 | 49.9 | 44.9 | 34.6 | 43.7 | 24.1 | 32.8 | 23.0 |
| BPA | 89.9* | 79.6 | 88.1 | 96.4 | 66.9 | 30.4 | 46.0 | 64.4 | 65.7 | 37.5 | 69.2 | 27.9 | 47.9 | 28.1 |
| LI-Boost-BPA | 93.0* | 86.1 | 92.1 | 98.4 | 77.3 | 39.4 | 53.7 | 73.8 | 75.0 | 40.6 | 81.4 | 34.7 | 57.1 | 43.3 |
| VDC | 51.7 | 58.6 | 67.0 | 65.6 | 52.1 | 97.5* | 55.2 | 59.3 | 71.5 | 38.2 | 41.8 | 28.8 | 35.8 | 28.4 |
| LI-Boost-VDC | 61.3 | 65.8 | 73.4 | 72.9 | 62.0 | 96.7* | 66.7 | 68.9 | 76.8 | 39.4 | 52.0 | 33.9 | 41.9 | 38.9 |
| FPR | 43.2 | 51.8 | 57.0 | 57.4 | 43.5 | 98.2* | 45.8 | 49.7 | 61.3 | 35.4 | 33.7 | 24.8 | 30.4 | 22.0 |
| LI-Boost-FPR | 53.5 | 57.8 | 63.7 | 63.5 | 54.6 | 96.8* | 58.1 | 60.4 | 68.3 | 36.7 | 43.3 | 27.9 | 34.8 | 29.9 |

formance on ResNet-50, underscoring the advantage of increasing local invariance to strengthen adversarial perturbations. Regarding black-box performance, MI-FGSM exhibits the lowest transferability among the baseline methods, whereas VMI-FGSM, PGN and MUMODIG demonstrate considerably stronger attack capabilities. Notably, LI-Boost consistently boosts the performance of these attacks across both CNNs and emerging ViT architectures. On average, the attack success rates show significant improvement, with the increases of 12.2%, 12.0%, 2.6% and 10.0% for MI-FGSM, VMI-FGSM, PGN and MUMODIG, respectively. These consistent and substantial performance gains highlight the effectiveness and generalizability of LI-Boost across diverse model architectures and defense strategies. Furthermore, even when facing robust defense mechanisms, LI-Boost significantly enhances the attack performance, revealing the limitations of existing defenses and raising new concerns about the robustness of models.

### 4.3 EVALUATION ON INPUT TRANSFORMATION-BASED ATTACKS

To assess the generality of LI-Boost, we integrate it with four prominent input transformation-based attacks, i.e., DIM, *Admix*, SIA and BSR. As shown in Tab. 2, LI-Boost significantly enhances the performance of white-box attacks, achieving near-perfect success rates of approximately 100.0%. This further corroborates the hypothesis that increasing local invariance strengthens adversarial attacks. Under black-box settings, LI-Boost consistently boosts the performance of various input transformation-based attacks. Overall, the integration of LI-Boost results in substantial performance gains over the baseline methods: an improvement of $5.4\% \sim 15.1\%$ for DIM, $8.9\% \sim 19.3\%$ for *Admix*, and $3.9\% \sim 20.5\%$ for SIA and $2.1\% \sim 17.5\%$ for BSR. Furthermore, attacks augmented with LI-Boost demonstrate superior robustness under various defense mechanisms. These significant improvements underscore the effectiveness of LI-Boost in enhancing adversarial transferability across diverse attack scenarios. Please refer to Appendix Tab. 9 for the results on other models.

### 4.4 EVALUATION ON MODEL-RELATED ATTACKS

To evaluate the efficacy of LI-Boost in model-related attacks, we integrate it with five prominent model-related attack methods, i.e., SGM, Linbp, BPA for CNNs and VDC, FPR for ViTs. The experimental results, presented in Tab. 3, demonstrate that attacks augmented with LI-Boost not only maintain high success rates in white-box settings but also achieve substantial improvements over the baseline methods in black-box scenarios: $14.1\% \sim 24.2\%$ for SGM, $5.6\% \sim 23.4\%$ for Linbp, $2.0\% \sim 10.4\%$ for BPA, $5.3\% \sim 11.5\%$ for VDC and $6.0\% \sim 12.3\%$ for FPR. These results highlight that LI-Boost significantly outperforms the baseline methods by considerable margins. Moreover, LI-Boost consistently enhances attack performance, achieving higher success rates across all evaluated defense strategies. These findings underscore the effectiveness of LI-Boost in augmenting adversarial attacks and suggest its potential as a robust approach for generating highly transferable adversarial examples.

Table 4: Attack success rates (%) on nine models and five defense approaches of various **advanced objective functions** w/wo LI-Boost. The adversarial examples are crafted on ResNet-50. * indicates the white-box model.

| Advanced Objective Functions | CNNs | | | | | ViTs | | | | Defenses | | | | |
|---|---|---|---|---|---|---|---|---|---|---|---|---|---|---|
| | RN-50 | Inc-v3 | MN-v3 | DN-121 | FSNet | ViT | PiT | Visformer | Swin | AT | HGD | RS | NRP | DiffPure |
| ILA | 90.0* | 29.0 | 37.9 | 42.4 | 22.4 | 8.2 | 15.4 | 20.8 | 27.1 | 33.4 | 13.9 | 21.6 | 20.0 | 11.4 |
| LI-Boost-ILA | 93.2* | 41.3 | 56.7 | 64.6 | 36.2 | 12.2 | 23.2 | 33.4 | 39.1 | 33.8 | 24.3 | 22.6 | 24.6 | 14.1 |
| FIA | 77.8* | 37.5 | 45.1 | 53.4 | 23.3 | 8.1 | 15.8 | 20.9 | 29.1 | 35.3 | 16.6 | 23.4 | 24.7 | 12.2 |
| LI-Boost-FIA | 89.6* | 53.6 | 65.1 | 76.2 | 42.7 | 13.9 | 25.5 | 37.7 | 45.4 | 36.7 | 32.8 | 25.2 | 30.8 | 15.4 |
| ILPD | 95.0* | 65.6 | 74.1 | 80.6 | 65.0 | 62.0 | 52.7 | 61.4 | 61.9 | 46.8 | 57.0 | 27.5 | 55.3 | 28.5 |
| LI-Boost-ILPD | 94.3* | 69.5 | 79.7 | 84.7 | 69.6 | 66.9 | 56.3 | 67.7 | 65.9 | 51.5 | 62.5 | 31.0 | 58.3 | 35.1 |
| BFA | 98.8* | 82.9 | 90.5 | 94.5 | 84.4 | 46.0 | 67.5 | 79.8 | 79.7 | 39.5 | 77.0 | 29.0 | 68.5 | 27.1 |
| LI-Boost-BFA | 98.7* | 86.8 | 92.6 | 96.0 | 87.9 | 53.8 | 72.1 | 84.8 | 83.8 | 42.3 | 83.7 | 36.3 | 74.5 | 44.4 |

Table 5: Attack success rates (%) on nine models and five defense approaches of various **ensemble attacks** w/wo LI-Boost. The adversarial examples are crafted on ResNet-50, Inc-v3, MobileNet-v3 and DenseNet-121. * indicates the white-box model.

| Ensemble Attacks | CNNs | | | | | ViTs | | | | Defenses | | | | |
|---|---|---|---|---|---|---|---|---|---|---|---|---|---|---|
| | RN-50 | Inc-v3 | MN-v3 | DN-121 | FSNet | ViT | PiT | Visformer | Swin | AT | HGD | RS | NRP | DiffPure |
| MI-FGSM | 95.4* | 99.8* | 99.3* | 100.0* | 67.8 | 39.3 | 53.7 | 66.6 | 68.5 | 37.2 | 66.6 | 27.1 | 44.7 | 24.6 |
| LI-Boost-MI | 97.9* | 100.0* | 99.6* | 100.0* | 85.6 | 58.4 | 71.0 | 83.2 | 83.9 | 39.9 | 85.3 | 34.5 | 58.3 | 40.9 |
| VMI-FGSM | 97.3* | 99.9* | 99.4* | 100.0* | 84.7 | 60.1 | 73.0 | 82.4 | 83.3 | 40.5 | 84.5 | 33.4 | 66.0 | 39.7 |
| LI-Boost-VMI | 99.3* | 100.0* | 99.7* | 100.0* | 93.1 | 73.7 | 84.5 | 91.8 | 92.4 | 45.0 | 94.0 | 42.6 | 83.6 | 58.2 |
| PGN | 98.8* | 100.0* | 99.6* | 100.0* | 94.6 | 81.2 | 88.7 | 94.1 | 94.1 | 54.9 | 95.9 | 58.4 | 90.0 | 71.0 |
| LI-Boost-PGN | 98.7* | 99.7* | 99.5* | 100.0* | 95.4 | 83.6 | 90.0 | 94.5 | 94.6 | 60.4 | 96.7 | 68.8 | 93.3 | 83.6 |
| MUMODIG | 99.6* | 99.8* | 99.8* | 100.0* | 97.2 | 84.2 | 92.3 | 97.1 | 96.3 | 46.4 | 98.0 | 40.4 | 82.4 | 52.8 |
| LI-Boost-MUMODIG | 99.6* | 99.8* | 99.8* | 100.0* | 98.1 | 89.1 | 95.0 | 98.2 | 97.5 | 51.9 | 98.8 | 51.1 | 90.6 | 72.8 |
| DIM | 97.8* | 99.9* | 99.6* | 100.0* | 86.1 | 61.5 | 74.5 | 84.4 | 84.7 | 39.9 | 87.6 | 31.9 | 60.4 | 38.2 |
| LI-Boost-DIM | 99.0* | 99.9* | 99.8* | 100.0* | 93.2 | 76.1 | 84.4 | 92.4 | 92.1 | 44.4 | 94.4 | 42.7 | 71.8 | 58.1 |
| *Admix* | 99.5* | 100.0* | 99.8* | 100.0* | 92.7 | 69.5 | 82.7 | 91.8 | 92.0 | 44.5 | 93.5 | 37.7 | 75.9 | 43.2 |
| LI-Boost-*Admix* | 99.4* | 100.0* | 100.0* | 100.0* | 96.1 | 83.1 | 89.6 | 95.6 | 95.4 | 51.0 | 95.0 | 50.5 | 85.5 | 70.2 |
| SIA | 99.8* | 100.0* | 100.0* | 100.0* | 98.0 | 82.3 | 93.6 | 97.9 | 97.3 | 44.6 | 98.6 | 37.5 | 78.7 | 46.1 |
| LI-Boost-SIA | 99.9* | 99.9* | 100.0* | 100.0* | 99.5 | 91.9 | 96.8 | 99.3 | 99.0 | 51.8 | 99.5 | 53.2 | 89.5 | 71.0 |
| BSR | 99.8* | 99.6* | 100.0* | 100.0* | 89.5 | 80.4 | 92.5 | 97.6 | 96.0 | 45.7 | 98.2 | 37.8 | 74.3 | 48.4 |
| LI-Boost-BSR | 99.9* | 99.9* | 100.0* | 100.0* | 99.3 | 90.6 | 95.8 | 99.2 | 98.7 | 52.4 | 99.5 | 53.4 | 86.2 | 74.4 |

## 4.5 EVALUATION ON ADVANCED OBJECTIVE FUNCTIONS

To validate the effectiveness of LI-Boost in advanced objective functions, we integrate our LI-Boost with ILA, FIA, ILPD and BFA. The results are presented in Tab. 4. As we can see from the table, under white-box setting, LI-Boost significantly improves the success rates of ILA and FIA by 3.2% and 11.8%, respectively, while maintaining performance of ILPD and BFA. For black-box settings, ILA exhibits the weakest performance among the three baseline methods, whereas FIA, ILPD and BFA demonstrate superior efficacy. Notably, LI-Boost substantially enhances the attack performance across both CNNs and ViTs. In particular, the magnitudes of improvement for ILA, FIA, ILPD and BFA are $3.2\% \sim 22.2\%$, $5.8\% \sim 22.8\%$, $3.6\% \sim 6.3\%$ and $1.5\% \sim 7.8\%$, respectively. Additionally, we evaluate the attack performance against five different defense mechanisms, where LI-Boost can still boost the baselines' performance. For instance, ILPD achieves an average success rate of 41.2% while LI-Boost-ILPD attains 45.0%. These performance improvements convincingly illustrate that LI-Boost can significantly boost the adversarial transferability.

## 4.6 EVALUATION ON ENSEMBLE ATTACK

To further validate the efficacy of our method, we adopt the ensemble attack as in MI-FGSM Dong et al. (2018), by fusing the logit outputs of diverse models. The adversarial examples are generated on RN-50, Inc-v3, MN-v3 and DN-121 using eight baselines w/wo LI-Boost and all ensemble models are assigned equal weights. As shown in Tab. 5, empirical results reveal that baseline methods consistently achieve enhanced adversarial transferability when integrated with LI-Boost. The augmented methods not only exhibit improved attack success rates in white-box scenarios but also demonstrate remarkable performance gains in black-box settings. Furthermore, comprehensive evaluations across five representative defense mechanisms highlight the robustness of our approach. These findings further highlight the effectiveness of LI-Boost in enhancing transferability.

## 4.7 ABLATION STUDIES

To gain deeper insights into LI-Boost, we conduct a series of ablation experiments to study the impact of hyper-parameters, i.e., the random sampling distribution, the number of sampled perturbations $N$, and the upper bound of translated pixels $k$. All the adversarial examples are generated on ResNet-50. The default setting is $N = 30$, $k = 6$, and Logarithmic distribution for sampling.

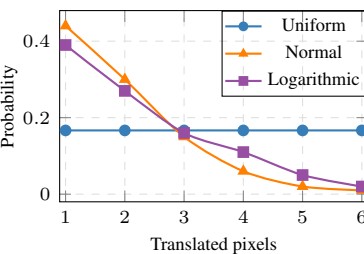
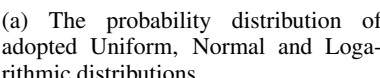

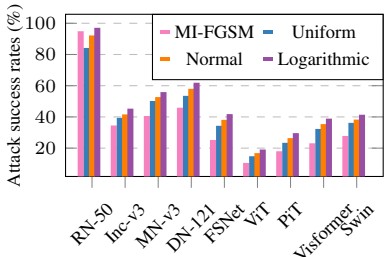

(a) The probability distribution of adopted Uniform, Normal and Logarithmic distributions.

(b) Attack success rates (%) of MI-FGSM and LI-Boost-MI with three distributions on eight models.

Figure 4: Ablation studies of various sampling distributions.

**On the sampling distribution.** Intuitively, local invariance within smaller neighborhoods holds greater significance than that in larger neighborhoods. Consequently, the choice of sampling distribution plays a critical role. To explore the impact of sampling distribution, we employ three distinct distributions as illustrated in Fig. 4a. As shown in Fig. 4b, Uniform distribution yields the weakest performance, as it fails to differentiate among translated pixels.

Nevertheless, it substantially surpasses MI-FGSM, highlighting the superiority of LI-Boost. Both Normal and Logarithmic distributions achieve better attack performance since they assign various levels of importance to different translated pixels. Logarithmic distribution achieves the best attack performance as it places suitable emphasis on smaller neighborhoods, which validates our hypothesis. Hence, we adopt Logarithmic distribution in our experiments. The details of sampling strategies are illustrated in Appendix Material A.4.

**On the number of sampled perturbations $N$.** We test LI-Boost-MI with various $N$ to analyze its impact on attack performance. As shown in Fig. 5a, the attack performance is significantly boosted with larger $N$ but exhibits diminishing returns beyond $N = 30$. Considering the growth of computational cost from gradient computations as shown in Eq. equation 5, we emperically select $N = 30$ in our experiments.

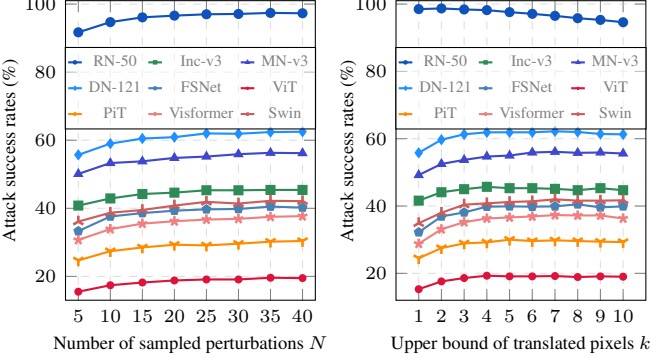

(a) The hyper-parameter $N$

(b) The hyper-parameter $k$

Figure 5: The attack success rates(%) on eight models with various hyper-parameters $N$ and $k$. The adversarial examples are generated by LI-Boost-MI on ResNet-50.

**On the upper bound of translated pixels $k$.** We conduct LI-Boost-MI using various $k$ to explore its impact on attack performance. Fig. 5b shows that $k = 1$ already surpasses MI-FGSM, demonstrating enhanced transferability. Performance peaks at around $k = 6$, highlighting the role of local invariance in robustness, while excessive $k$ values degrade as generating effective perturbations becomes more challenging. Thus, we select $k = 6$ to balance the white-box and transferable attack efficacy.

## 5 CONCLUSIONS

In this study, we introduce *local invariance* of adversarial perturbations and empirically demonstrate a positive correlation between the local invariance of adversarial perturbations on *a surrogate model* and their transferability *across diverse victim models*. Building on this insight, we propose LI-Boost, a novel method designed to enhance the local invariance of adversarial perturbations on a single model for better adversarial transferability. Through extensive experiments conducted on the standard ImageNet validation set, we validate the effectiveness of LI-Boost across a variety of transfer-based attacks, encompassing both CNNs, ViTs and various defense mechanisms. Our findings not only underscore the efficacy of the proposed approach but also provide valuable insights into potential avenues for advancing adversarial attack. We anticipate that this work will inspire further research in this direction.

ETHICS STATEMENT

This paper focuses on the transferability of adversarial examples. The dataset was curated following ethical guidelines to ensure that no sensitive information is included and minimize bias. The evaluation process aims to be transparent and reproducible, adhering to high standards of research integrity and ethical conduct. No personally identifiable data was collected or processed.

REPRODUCIBILITY STATEMENT

To ensure the reproducibility of our results, we have made considerable efforts to provide all necessary details and materials. Specifically, we have included the dataset in Section 4.1, the hardware setting in Section 4.1, provided the algorithm details in Algorithm 1 and Appendix A.4.

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

# A APPENDIX

## A.1 FURTHER DISCUSSION

Through the above experiments, we have validated that LI-Boost can significantly boost the adversarial transferability of various transfer-based attacks across different models and defense mechanisms. To further substantiate our hypothesis that enhancing local invariance improves adversarial transferability, we quantify the local invariance of six transfer-based attacks w/wo LI-Boost, namely MI-FGSM, DIM, BSR, BPA, FPR and ILA.

As shown in Tab. 6, the results are consistent with that in Fig. 2 in main paper, showing that higher local invariance correlates with improved adversarial transferability. Furthermore, LI-Boost effectively increases the local invariance of generated adversarial perturbations, thereby concurrently improving adversarial transferability. It validates our motivation that LI-Boost can boost the local invariance to enhance the adversarial transferability.

Table 6: Average attack success rates of nine models (%) of various attacks and local invariance ($k = 6$) on ResNet-50 w/wo LI-Boost. The adversarial examples are generated on ResNet-50.

| LI-Boost | MI-FGSM | DIM | BSR | BPA | FPR | ILA |
|---|---|---|---|---|---|---|
| ✗ | 33.4/0.24 | 48.9/0.31 | 83.7/0.48 | 67.0/0.70 | 56.4/0.30 | 30.4/0.15 |
| ✓ | **45.3/0.41** | **60.5/0.51** | **91.0/0.81** | **74.2/0.88** | **64.1/0.42** | **41.7/0.34** |

## A.2 LIMITATIONS

Although we have experimentally verified that perturbations with enhanced local invariance can improve the adversarial transferability, there is still a lack of theoretical analysis on the relationship between local invariance and adversarial transferability. In future work, we will continue exploring from a theoretical perspective to provide valuable insights into adversarial attacks.

## A.3 THE USE OF LARGE LANGUAGE MODELS

In the preparation of this work, we used an Large Language Model (LLM) solely for grammatical improvement. The models are not involved in generating technical content, ideas, experimental design, or results interpretation.

It is important to note that all scientific contributions, including conceptualization, analysis and conclusions, are entirely the work of the authors.

## A.4 SAMPLING DISTRIBUTIONS

In this section, we detail the three sampling distributions for pixel translation employed in our study: uniform, normal, and logarithmic. For the sake of simplicity, the random variable for these distributions is defined as the number of translated pixels. Given the actual number of translated pixels $x_p$, upperbound $k$, the probability mass function are as follows:

**Uniform:**

$$P_{\text{uniform}}(X = x_p; k) = \begin{cases} \frac{1}{k}, & \text{for } x_p \in \{1, 2, \ldots, k\} \\ 0, & \text{otherwise} \end{cases} \tag{6}$$

**Normal:**

$$P_{\text{normal}}(X = x_p; \mu = 0, \sigma, k) = \begin{cases} \frac{\exp(-\frac{x_p^2}{2\sigma^2})}{Z_{\text{normal}}}, & \text{for } x_p \in \{1, 2, \ldots, k\} \\ 0, & \text{otherwise} \end{cases} \tag{7}$$

where $Z_{\text{normal}}$ is the normalization constant, given by:

$$Z_{\text{normal}} = \sum_{i=1}^{k} \exp(-\frac{i^2}{2\sigma^2}) \tag{8}$$

**Logarithmic:**

$$P_{\text{logarithmic}}(X = x_p; k) = \begin{cases} \frac{\ln(\frac{k+1}{x_p})}{Z_{\text{logarithmic}}}, & \text{for } x_p \in \{1, 2, \ldots, k\} \\ 0, & \text{otherwise} \end{cases} \tag{9}$$

where $Z_{\text{logarithmic}}$ is the normalization constant, defined as:

$$Z_{\text{logarithmic}} = \sum_{i=1}^{k} \ln(\frac{k+1}{i}) \tag{10}$$

## A.5 VISULIZATION AND APPLICATION IN PHYSICAL SCENARIO

This section presents adversarial examples generated by three methods—MI-FGSM, BSR, and FPR—each augmented with our LI-Boost enhancement. To evaluate their real-world efficacy, we deploy these attacks against the Baidu Cloud API. As shown in Fig. 6, each pair of images consists of an benign, correctly classified image (top) and its corresponding adversarial example (bottom) crafted by LI-Boost-enhanced MI-FGSM, BSR, and FPR, respectively.

## A.6 PARAMETER SETTINGS

In this section, we provide the detailed parameter settings for the baseline attacks employed in our work. These settings are consistent with the corresponding papers to ensure fair and comprehensive evaluations. Below, we delineate the hyperparameters for each category of baseline methods in Tab. 7. All defense models are pre-trained on the ImageNet dataset and evaluated on a single model.

AT and HGD adopt the official models provided in the corresponding papers. RS utilizes the defense model ResNet-50 with a noise level of $0.5$. For NRP and DiffPure, we choose ResNet-101 as the target classifier.

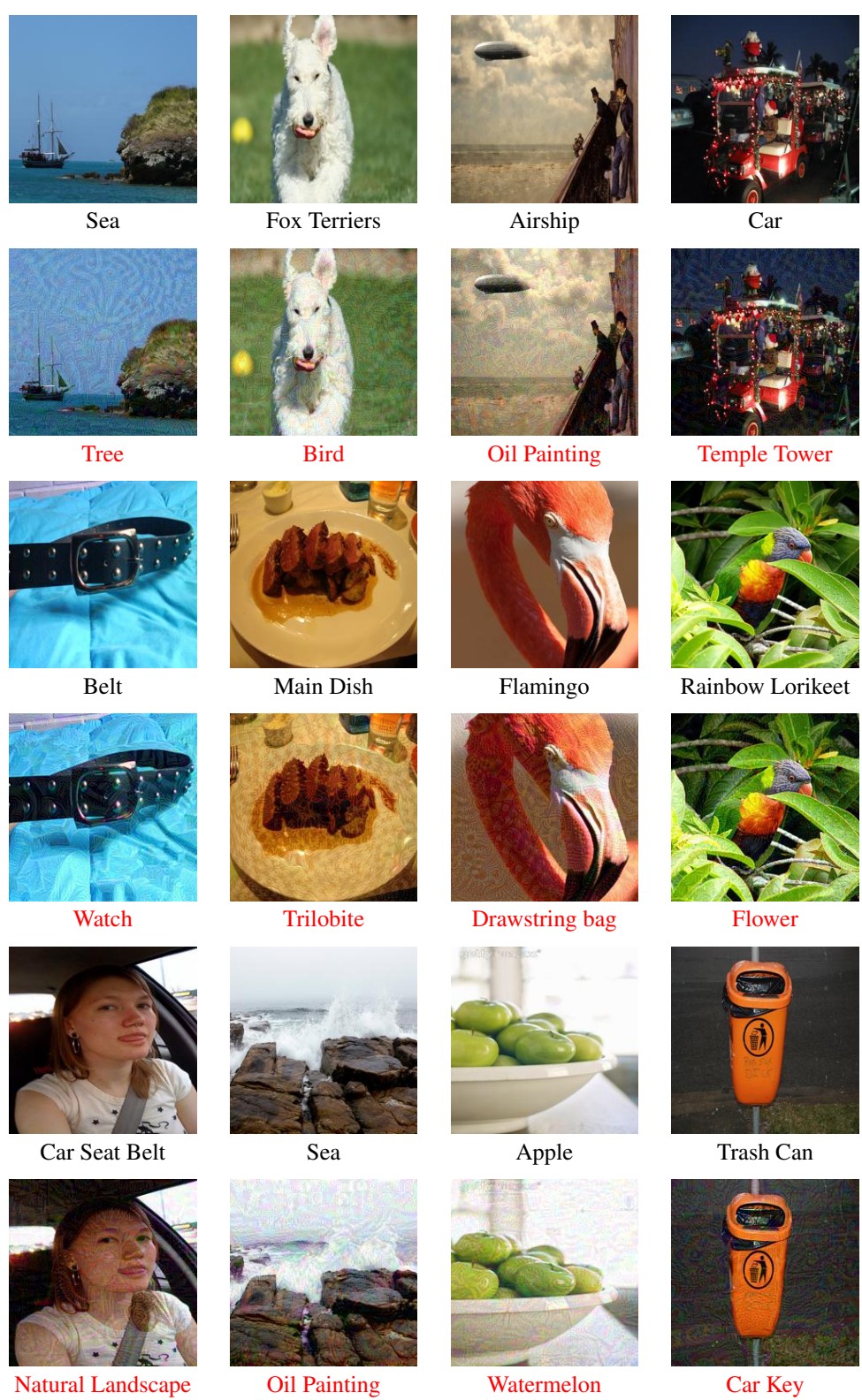

Figure 6: Visulization of benign images and their adversarial counterparts, along with their corresponding classifications.

| Method | Parameters |
|---|---|
| **Gradient-based Attacks** | |
| MI-FGSM( Dong et al. (2018)) | perturbation budget $\epsilon = 16/255$, 
 number of iterations $T = 10$, 
 step size $\alpha = \epsilon/T = 1.6/255$, 
 decay factor $\mu = 1.0$ |
| VMI-FGSM( Wang & He (2021)) | number of sampled examples $N_s = 20$, 
 upper bound of neighborhood $\zeta = 1.5$ |
| PGN( Ge et al. (2023)) | number of sampled examples $N_s = 20$, 
 balanced coefficient $c_b = 0.5$, 
 upper bound of neighborhood $\zeta = 3.0 \times \epsilon$ |
| MUMODIG( Ren et al. (2025b)) | position factor $\lambda_p = 0.65$, 
 region number = $N_R = 2$, 
 interpolation point number $N_T = 1$, 
 number of sampled baselines $N_B = 1$, 
 number of sampled transformations $N_T = 6$ |
| **Input Transformation-based Attacks** | |
| DIM( Xie et al. (2019)) | resize rate $r = 1.1$, 
 diversity probability $p_{di} = 0.5$ |
| *Admix*( Wang et al. (2021a)) | number of scaled copies $m_1 = 5$, 
 number of admixed images $m_2 = 3$, 
 admix strength $\eta = 0.2$ |
| SIA( Wang et al. (2023b)) | number of blocks $s = 3$, 
 number of transformed images $N_t = 20$ |
| BSR( Wang et al. (2024a)) | number of blocks $s = 3$, 
 number of shuffled images $N_u = 20$, 
 range of rotation angles $\tau = 24°$ |
| **Model-related Attacks** | |
| SGM( Wu et al. (2020)) | residual gradient decay $\gamma = 0.5$ |
| Linbp( Guo et al. (2020)) | number of iterations $T = 300$, 
 the first layer to be modified is the first residual unit in the third meta block. |
| BPA( Wang et al. (2023a)) | temperature coefficient $c_t = 10$, 
 the first layer to be modified is the first residual unit in the third meta block. |
| VDC( Zhang et al. (2024a)) | patch size $P_s = 16$, 
 scale factor $s_f = 0.5$, 
 residual gradient decay $\gamma = 0.5$ |
| FPR( Ren et al. (2025a)) | diversity factor $d_f = 25$, 
 scale factor $s_f = 0.8$, 
 attenuation factor $a_f = 0.3$, 
 index set of diversified blocks $I = [0, 1, 4, 9, 11]$ |
| **Advanced Objective Functions** | |
| ILA( Huang et al. (2019)) | coefficient $c = 1.0$ |
| FIA( Wang et al. (2021c)) | drop probability $p_{dr} = 0.3$, 
 number of aggregated gradients $N_a = 30$, 
 the target layer to attack is the last layer of the second block. |
| ILPD( Li et al. (2023)) | number of iterations $T = 100$, 
 noise size $\sigma = 0.05$, 
 coefficient $c = 0.1$, 
 step size $\alpha = 1/255$, 
 the target layer to attack is the third building block of the second ResNet meta layer. |
| BFA( Wang et al. (2024b)) | perturbation mask size $s_{mask} = 28$, 
 number of fitting iteration steps $T = 30$, 
 the target layer to attack is the last layer of the second block |

Table 7: Hyper-parameters for various transfer-based attack baselines.

| Surrogate Model | Attack | RN-50 | Inc-v3 | MN-v3 | DN-121 | FSNet | ViT | PiT | Visformer | Swin |
|---|---|---|---|---|---|---|---|---|---|---|
| Inc-v3 | MI-FGSM | 34.1 | 99.0* | 46.7 | 50.6 | 25.7 | 14.2 | 20.5 | 26.6 | 31.8 |
| | LI-Boost-MI | 43.8 | 99.1* | 56.9 | 62.6 | 37.3 | 20.8 | 26.4 | 35.0 | 40.9 |
| | VMI-FGSM | 50.0 | 99.4* | 60.3 | 66.3 | 41.6 | 25.5 | 32.7 | 39.7 | 44.8 |
| | LI-Boost-VMI | 52.5 | 99.5* | 62.0 | 69.1 | 44.2 | 27.0 | 35.1 | 42.6 | 48.0 |
| | PGN | 63.1 | 100.0* | 75.7 | 81.2 | 55.7 | 55.7 | 43.7 | 51.7 | 57.3 |
| | LI-Boost-PGN | 77.3 | 100.0* | 83.8 | 89.7 | 70.8 | 47.4 | 57.4 | 66.6 | 70.4 |
| | MUMODIG | 65.1 | 99.5* | 76.6 | 81.7 | 57.0 | 34.0 | 42.7 | 53.9 | 58.2 |
| | LI-Boost-MUMODIG | 89.4 | 99.9* | 92.1 | 96.2 | 85.0 | 62.4 | 73.5 | 83.4 | 81.8 |
| MN-v3 | MI-FGSM | 41.7 | 50.6 | 100.0* | 60.0 | 31.4 | 17.9 | 26.6 | 36.9 | 42.7 |
| | LI-Boost-MI | 65.6 | 67.6 | 100.0* | 80.7 | 53.3 | 33.0 | 46.1 | 61.1 | 65.7 |
| | VMI-FGSM | 67.5 | 73.0 | 99.9* | 80.7 | 57.6 | 37.2 | 51.6 | 64.2 | 69.7 |
| | LI-Boost-VMI | 75.5 | 79.0 | 99.9* | 87.2 | 66.0 | 44.5 | 58.6 | 71.7 | 77.2 |
| | PGN | 80.3 | 86.5 | 100.0* | 92.1 | 70.9 | 49.6 | 64.7 | 76.4 | 82.5 |
| | LI-Boost-PGN | 84.9 | 89.8 | 100.0* | 94.5 | 78.9 | 59.1 | 71.9 | 82.0 | 86.2 |
| | MUMODIG | 83.5 | 88.7 | 99.9* | 93.3 | 75.0 | 51.8 | 68.7 | 81.0 | 82.7 |
| | LI-Boost-MUMODIG | 90.1 | 92.0 | 100.0* | 96.0 | 84.6 | 68.9 | 81.0 | 88.8 | 90.2 |
| DN-121 | MI-FGSM | 67.1 | 61.5 | 71.5 | 100.0* | 49.6 | 24.3 | 33.6 | 47.9 | 50.2 |
| | LI-Boost-MI | 82.2 | 74.9 | 84.8 | 100.0* | 70.1 | 35.8 | 47.1 | 66.1 | 66.3 |
| | VMI-FGSM | 84.7 | 79.7 | 86.0 | 100.0* | 72.3 | 42.7 | 54.6 | 69.6 | 70.5 |
| | LI-Boost-VMI | 90.9 | 86.5 | 91.1 | 100.0* | 80.7 | 50.1 | 63.6 | 78.1 | 78.6 |
| | PGN | 94.1 | 93.3 | 95.2 | 100.0* | 86.9 | 60.0 | 72.4 | 84.8 | 85.1 |
| | LI-Boost-PGN | 95.0 | 94.1 | 95.2 | 100.0* | 88.6 | 64.4 | 75.5 | 86.4 | 86.9 |
| | MUMODIG | 95.2 | 93.6 | 95.0 | 100.0* | 86.7 | 55.7 | 69.0 | 84.5 | 82.2 |
| | LI-Boost-MUMODIG | 97.2 | 96.5 | 97.1 | 100.0* | 92.8 | 70.7 | 80.7 | 92.0 | 89.4 |
| FSNet | MI-FGSM | 44.8 | 42.5 | 51.6 | 53.4 | 97.6* | 20.0 | 31.8 | 39.9 | 47.5 |
| | LI-Boost-MI | 65.4 | 71.5 | 71.2 | 75.4 | 99.4* | 37.8 | 54.8 | 67.2 | 69.6 |
| | VMI-FGSM | 69.7 | 62.8 | 69.3 | 73.0 | 98.6* | 44.0 | 57.6 | 66.1 | 70.5 |
| | LI-Boost-VMI | 88.4 | 80.5 | 85.8 | 89.6 | 99.8* | 65.6 | 79.1 | 86.1 | 88.2 |
| | PGN | 93.7 | 89.9 | 92.0 | 94.5 | 99.5* | 78.6 | 88.4 | 92.1 | 93.1 |
| | LI-Boost-PGN | 94.6 | 92.2 | 93.4 | 95.6 | 99.2* | 83.1 | 90.7 | 93.4 | 93.7 |
| | MUMODIG | 88.3 | 81.5 | 86.0 | 89.9 | 99.0* | 63.2 | 79.2 | 85.7 | 85.8 |
| | LI-Boost-MUMODIG | 92.4 | 85.9 | 89.8 | 93.3 | 99.0* | 74.4 | 85.7 | 90.7 | 90.8 |
| ViT | MI-FGSM | 43.7 | 51.3 | 57.8 | 57.2 | 43.4 | 98.2* | 45.6 | 49.3 | 61.5 |
| | LI-Boost-MI | 53.2 | 57.8 | 63.2 | 64.7 | 53.4 | 97.0* | 58.1 | 60.1 | 68.4 |
| | VMI-FGSM | 55.7 | 61.8 | 66.9 | 66.1 | 58.3 | 99.1* | 62.1 | 64.0 | 73.3 |
| | LI-Boost-VMI | 61.6 | 67.8 | 72.1 | 71.7 | 65.6 | 99.6* | 68.7 | 69.5 | 77.5 |
| | PGN | 76.3 | 78.9 | 83.7 | 83.1 | 78.5 | 99.2* | 83.4 | 83.2 | 87.6 |
| | LI-Boost-PGN | 78.2 | 80.8 | 84.6 | 84.9 | 80.7 | 99.1* | 84.7 | 84.7 | 88.5 |
| | MUMODIG | 70.9 | 74.9 | 78.2 | 77.1 | 73.1 | 95.8* | 77.8 | 78.1 | 80.9 |
| | LI-Boost-MUMODIG | 77.7 | 78.1 | 82.6 | 83.0 | 79.5 | 98.1* | 83.7 | 84.0 | 86.3 |
| PiT | MI-FGSM | 44.3 | 48.5 | 57.0 | 54.4 | 41.3 | 30.6 | 97.9* | 50.0 | 53.5 |
| | LI-Boost-MI | 56.8 | 56.1 | 67.0 | 64.5 | 54.4 | 45.0 | 98.0* | 64.6 | 67.6 |
| | VMI-FGSM | 61.6 | 62.1 | 69.9 | 67.8 | 61.7 | 52.3 | 97.9* | 69.5 | 72.0 |
| | LI-Boost-VMI | 69.7 | 70.4 | 76.6 | 75.7 | 70.9 | 60.4 | 99.3* | 77.1 | 78.5 |
| | PGN | 78.9 | 79.4 | 83.5 | 82.7 | 80.1 | 76.3 | 97.5* | 84.5 | 85.3 |
| | LI-Boost-PGN | 79.5 | 80.5 | 83.7 | 82.9 | 80.5 | 77.5 | 96.7* | 84.6 | 85.2 |
| | MUMODIG | 76.2 | 75.7 | 80.9 | 79.7 | 77.8 | 69.8 | 96.9* | 82.8 | 83.7 |
| | LI-Boost-MUMODIG | 81.7 | 79.9 | 85.3 | 84.7 | 83.0 | 78.6 | 98.3* | 88.2 | 88.2 |
| Visformer | MI-FGSM | 52.4 | 52.5 | 65.6 | 63.5 | 53.6 | 32.8 | 52.0 | 98.6* | 64.0 |
| | LI-Boost-MI | 68.4 | 64.0 | 77.7 | 77.4 | 71.3 | 51.1 | 69.8 | 98.0* | 78.4 |
| | VMI-FGSM | 73.7 | 71.0 | 80.6 | 80.3 | 76.7 | 59.8 | 76.7 | 98.8* | 82.8 |
| | LI-Boost-VMI | 77.8 | 75.3 | 82.9 | 82.7 | 80.1 | 66.2 | 80.4 | 98.7* | 85.5 |
| | PGN | 88.6 | 87.5 | 91.5 | 92.4 | 90.0 | 83.3 | 90.9 | 98.7* | 92.7 |
| | LI-Boost-PGN | 89.1 | 88.3 | 91.3 | 92.7 | 89.5 | 84.3 | 90.9 | 98.6* | 92.5 |
| | MUMODIG | 88.8 | 85.8 | 91.8 | 91.8 | 89.9 | 76.2 | 90.5 | 99.1* | 92.4 |
| | LI-Boost-MUMODIG | 90.9 | 88.7 | 92.3 | 93.3 | 92.0 | 82.6 | 90.6 | 99.3* | 93.9 |
| Swin | MI-FGSM | 32.8 | 36.9 | 50.0 | 44.2 | 33.0 | 22.2 | 30.5 | 38.9 | 98.1* |
| | LI-Boost-MI | 59.0 | 55.0 | 73.6 | 68.7 | 59.8 | 44.5 | 59.2 | 69.3 | 99.4* |
| | VMI-FGSM | 57.4 | 58.5 | 71.6 | 66.6 | 69.1 | 51.0 | 61.9 | 68.9 | 98.7* |
| | LI-Boost-VMI | 76.3 | 76.7 | 87.8 | 84.4 | 81.4 | 71.3 | 81.4 | 87.3 | 100.0* |
| | PGN | 85.5 | 86.9 | 93.5 | 91.3 | 89.0 | 85.7 | 90.0 | 92.7 | 99.7* |
| | LI-Boost-PGN | 87.4 | 88.4 | 93.5 | 92.6 | 90.1 | 87.0 | 91.3 | 93.3 | 99.6* |
| | MUMODIG | 80.8 | 80.3 | 88.9 | 86.7 | 84.0 | 69.5 | 84.5 | 87.9 | 99.2* |
| | LI-Boost-MUMODIG | 87.4 | 86.0 | 92.9 | 91.7 | 89.9 | 81.9 | 90.6 | 93.2 | 99.8* |

Table 8: Attack success rates (%) of **gradient-based attacks** w/wo LI-Boost on nine models. The adversarial examples are crafted on Inc-v3, MN-v3, DN-121, FSNet, ViT, PiT, Visformer, and Swin respectively. * indicates white-box model.

| Surrogate Model | Attack | RN-50 | Inc-v3 | MN-v3 | DN-121 | FSNet | ViT | PiT | Visformer | Swin |
|---|---|---|---|---|---|---|---|---|---|---|
| **Inc-v3** | DIM | 46.0 | 99.0* | 58.6 | 65.0 | 39.2 | 22.0 | 29.3 | 36.3 | 41.6 |
| | LI-Boost-DIM | **55.6** | **99.6*** | **67.1** | **74.1** | **50.0** | **29.5** | **35.6** | **46.3** | **51.8** |
| | *Admix* | 56.3 | **99.9*** | 68.3 | 75.4 | 45.5 | 25.3 | 33.8 | 43.6 | 48.7 |
| | LI-Boost-*Admix* | **79.8** | 99.8* | **84.9** | **91.7** | **62.7** | **49.0** | **52.6** | **67.0** | **70.0** |
| | SIA | 77.5 | 99.8* | 88.0 | 90.6 | 66.9 | 39.3 | 52.9 | 66.0 | 68.7 |
| | LI-Boost-SIA | **91.3** | **99.9*** | **96.4** | **97.9** | **85.8** | **61.7** | **72.3** | **84.5** | **85.9** |
| | BSR | 78.7 | 99.8* | 88.8 | 92.3 | 69.5 | 43.2 | 54.9 | 68.5 | 70.6 |
| | LI-Boost-BSR | **90.7** | **100.0*** | **96.0** | **98.3** | **86.2** | **61.8** | **70.1** | **84.7** | **85.5** |
| **MN-v3** | DIM | 64.7 | 74.1 | 100.0* | 81.9 | 54.4 | 36.4 | 50.0 | 62.0 | 66.8 |
| | LI-Boost-DIM | **80.2** | **83.0** | 100.0* | **91.0** | **71.4** | **51.7** | **63.3** | **77.2** | **80.4** |
| | *Admix* | 70.9 | 75.8 | 100.0* | 85.3 | 58.1 | 36.5 | 53.0 | 66.8 | 71.9 |
| | LI-Boost-*Admix* | **85.9** | **88.6** | 100.0* | **94.3** | **76.5** | **59.1** | **70.5** | **83.4** | **85.2** |
| | SIA | 82.2 | 82.6 | 100.0* | 92.6 | 71.3 | 46.0 | 65.5 | 79.4 | 83.1 |
| | LI-Boost-SIA | **92.7** | **90.4** | 100.0* | **97.4** | **84.8** | **63.6** | **79.4** | **90.5** | **91.9** |
| | BSR | 88.7 | 89.2 | 100.0* | 96.0 | 79.5 | 57.2 | 76.5 | 85.7 | 87.3 |
| | LI-Boost-BSR | **93.8** | **92.8** | 100.0* | **98.3** | **87.1** | **69.2** | **81.7** | **91.6** | **92.0** |
| **DN-121** | DIM | 82.5 | 80.7 | 84.6 | 100.0* | 69.0 | 38.6 | 49.7 | 65.5 | 65.2 |
| | LI-Boost-DIM | **90.9** | **87.6** | **92.0** | 100.0* | **81.4** | **50.8** | **60.3** | **79.0** | **77.0** |
| | *Admix* | 91.3 | 87.5 | 91.3 | 100.0* | 77.8 | 45.2 | 58.7 | 75.9 | 74.5 |
| | LI-Boost-*Admix* | **95.8** | **94.7** | **96.7** | 100.0* | **88.8** | **67.0** | **71.0** | **87.8** | **86.3** |
| | SIA | 98.2 | 93.9 | 98.5 | 100.0* | 90.7 | 58.0 | 74.8 | 90.3 | 87.6 |
| | LI-Boost-SIA | **99.2** | **97.0** | **99.3** | 100.0* | **97.0** | **71.9** | **83.5** | **95.6** | **93.5** |
| | BSR | 97.4 | 94.6 | 97.7 | 100.0* | 90.1 | 60.9 | 75.7 | 89.5 | 86.3 |
| | LI-Boost-BSR | **98.5** | **96.5** | **98.7** | 100.0* | **94.2** | **68.1** | **78.1** | **93.5** | **90.8** |
| **FSNet** | DIM | 44.1 | 38.5 | 52.0 | 53.4 | 97.7* | 19.5 | 31.4 | 39.8 | 47.1 |
| | LI-Boost-DIM | **71.3** | **56.9** | **71.3** | **75.3** | **99.4*** | **37.8** | **54.8** | **67.6** | **70.1** |
| | *Admix* | 82.0 | 72.1 | 79.6 | 84.5 | **99.9*** | 52.7 | 69.0 | 79.5 | 81.9 |
| | LI-Boost-*Admix* | **90.8** | **82.0** | **88.9** | **92.0** | 99.8* | **69.3** | **81.0** | **88.6** | **89.5** |
| | SIA | 93.9 | 82.5 | 92.3 | 93.5 | 99.7* | 61.6 | 83.7 | 90.5 | 92.2 |
| | LI-Boost-SIA | **97.9** | **88.1** | **97.1** | **97.6** | **99.8*** | **78.8** | **92.3** | **96.4** | **96.7** |
| | BSR | 95.9 | 87.1 | 95.0 | 96.2 | 99.4* | 69.1 | 88.7 | 93.5 | 92.7 |
| | LI-Boost-BSR | **98.1** | **92.1** | **97.7** | **98.4** | **99.6*** | **80.3** | **92.8** | **96.6** | **96.4** |
| **ViT** | DIM | 55.4 | 61.9 | 64.7 | 65.4 | 58.6 | 93.2* | 62.3 | 62.7 | 68.3 |
| | LI-Boost-DIM | **64.0** | **66.7** | **71.8** | **72.0** | **70.8** | **96.8*** | **72.1** | **72.4** | **76.2** |
| | *Admix* | 61.2 | 66.9 | 72.2 | 71.9 | 63.2 | **99.3*** | 67.5 | 69.7 | 80.7 |
| | LI-Boost-*Admix* | **72.4** | **73.3** | **81.4** | **81.2** | **73.7** | 99.2* | **77.5** | **80.3** | **85.5** |
| | SIA | 82.3 | 80.2 | 88.4 | 86.7 | 82.9 | 99.2* | 88.3 | 87.7 | 90.8 |
| | LI-Boost-SIA | **88.8** | **84.8** | **92.7** | **91.7** | **89.6** | **99.7*** | **93.1** | **93.3** | **94.7** |
| | BSR | 85.9 | 85.1 | 89.8 | 89.1 | 86.6 | 96.1* | 90.6 | 89.9 | 90.4 |
| | LI-Boost-BSR | **89.7** | **87.4** | **93.0** | **92.2** | **90.5** | **97.5*** | **93.4** | **93.3** | **93.1** |
| **PiT** | DIM | 60.1 | 63.0 | 69.5 | 68.1 | 62.2 | 52.6 | 95.5* | 69.8 | 71.9 |
| | LI-Boost-DIM | **69.1** | **68.4** | **76.2** | **74.8** | **73.8** | **65.2** | **98.4*** | **78.4** | **79.2** |
| | *Admix* | 60.4 | 57.7 | 69.1 | 66.8 | 61.3 | 45.6 | 98.4* | 67.9 | 71.2 |
| | LI-Boost-*Admix* | **71.6** | **65.0** | **78.2** | **76.0** | **73.8** | **61.3** | **98.6*** | **79.1** | **81.0** |
| | SIA | 87.7 | 79.2 | 91.1 | 89.0 | 87.6 | 78.7 | 99.8* | 93.1 | 93.7 |
| | LI-Boost-SIA | **92.8** | **85.6** | **95.1** | **93.6** | **93.5** | **89.6** | **99.9*** | **97.0** | **97.0** |
| | BSR | 88.4 | 84.5 | 92.4 | 90.8 | 89.9 | 81.2 | 99.2* | 93.8 | 94.1 |
| | LI-Boost-BSR | **91.0** | **86.9** | **94.1** | **93.3** | **92.3** | **88.1** | **99.4*** | **95.5** | **95.9** |
| **Visformer** | DIM | 73.3 | 71.8 | 80.9 | 81.1 | 75.4 | 58.9 | 76.4 | 97.9* | 80.9 |
| | LI-Boost-DIM | **79.5** | **76.5** | **85.1** | **86.2** | **84.5** | **68.6** | **83.2** | **98.6*** | **86.7** |
| | *Admix* | 77.4 | 73.5 | 84.3 | 83.2 | 78.9 | 58.6 | 80.3 | **99.0*** | 86.3 |
| | LI-Boost-*Admix* | **84.4** | **79.7** | **89.2** | **89.1** | **86.2** | **72.2** | **86.9** | 98.8* | **90.4** |
| | SIA | 92.6 | 83.5 | 94.8 | 94.2 | 92.9 | 75.9 | 93.4 | 99.8* | 96.0 |
| | LI-Boost-SIA | **95.3** | **88.2** | **96.9** | **96.9** | **96.8** | **85.5** | **96.1** | **99.9*** | **97.9** |
| | BSR | 95.1 | 90.7 | 96.5 | 97.0 | 95.4 | 81.1 | 95.5 | **99.8*** | 96.8 |
| | LI-Boost-BSR | **96.4** | **92.3** | **97.5** | **98.0** | **96.9** | **87.4** | **96.9** | **99.8*** | **97.9** |
| **Swin** | DIM | 67.2 | 69.2 | 79.3 | 76.1 | 71.4 | 56.7 | 72.7 | 77.1 | 98.6* |
| | LI-Boost-DIM | **79.4** | **77.0** | **87.5** | **85.2** | **84.5** | **70.8** | **83.8** | **87.9** | **99.6*** |
| | *Admix* | 47.5 | 42.3 | 63.7 | 57.3 | 47.7 | 33.6 | 45.6 | 56.5 | 99.3* |
| | LI-Boost-*Admix* | **72.8** | **61.8** | **84.1** | **80.2** | **74.3** | **57.6** | **74.4** | **81.9** | **99.4*** |
| | SIA | 83.2 | 73.9 | 92.7 | 87.3 | 84.8 | 66.5 | 85.0 | 90.8 | **99.9*** |
| | LI-Boost-SIA | **92.8** | **84.1** | **97.6** | **95.2** | **95.5** | **82.6** | **94.6** | **96.9** | **99.9*** |
| | BSR | 92.4 | 87.5 | 96.8 | 95.2 | 93.7 | 77.7 | 94.9 | 95.9 | 99.4* |
| | LI-Boost-BSR | **96.1** | **91.5** | **98.3** | **97.6** | **97.0** | **87.4** | **97.5** | **98.0** | **99.7*** |

Table 9: Attack success rates (%) of **input transformation-based attacks** w/wo LI-Boost on nine models. The adversarial examples are crafted on Inc-v3, MN-v3, DN-121, FSNet, ViT, PiT, Visformer, and Swin respectively. * indicates white-box model.

