# OpenReview forum: "Boosting the Local Invariance for Better Adversarial Transferability"
_ICLR.cc/2026/Conference — ICLR 2026 Conference Withdrawn Submission_

### Official Review · Reviewer_8EuR · 2025-10-21

**Soundness:** 3
**Presentation:** 2
**Contribution:** 2
**Rating:** 4
**Confidence:** 4

**Summary:**

This paper addresses the transferability of adversarial examples through a novel perspective: local invariance. The authors find that existing adversarial perturbations often lose their effectiveness after translation, while clean images can still be correctly recognized after translation. Based on this observation, the authors propose a highly versatile method, LI-Boost, which introduces random translation perturbations and samples the average gradient during adversarial example generation to enhance local invariance, significantly improving the attack success rate across models (CNN, ViT) and defense methods. This method can be flexibly embedded in a variety of existing attack methods (such as MI-FGSM, DIM, and SGM). Extensive experiments on the ImageNet dataset demonstrate that it significantly improves transfer attack performance in both white-box and black-box settings.

**Strengths:**

1. Novel perspective and clear motivation: This paper systematically proposes and verifies the positive correlation between local invariance and adversarial transferability for the first time, which is thought-provoking.

2. General and easily integrated: LI-Boost is a plug-in boosting method that can be seamlessly integrated into gradient-based, input transformation-based, model structure-based, objective function-based, and ensemble-based attacks.

3. Comprehensive and convincing experiments: The effectiveness is verified on multiple model architectures (five CNNs + four ViTs) and five mainstream defense methods.

**Weaknesses:**

1. Computational Overhead. The proposed method is based on the random sampling. The number of random sampling is set to 30 in the final setting, meaning 30 times the computational overhead compared with the baseline method, which is not effective.

2. The advantage of Translation operation. Compared to translation operations, there are some other mutation operations., such as random mask, flip, rotation and other operations. I am curious whether translation operations are better than these mutation operations.

3. Compared with the selection of translation operation, can we design some kind of algorithm to optimize it instead of the current random selection? Can this improve efficiency?

**Questions:**

See the weakness

---

### Official Review · Reviewer_zEwH · 2025-10-25

**Soundness:** 2
**Presentation:** 2
**Contribution:** 2
**Rating:** 2
**Confidence:** 4

**Summary:**

This work introduces the connection between the local invariance of adversarial perturbations and the transferability of adversarial examples across ML models. The authors claim a positive correlation between local invariance of perturbations and their transferability, empirically exploring why adversarial examples generated by existing transfer-based methods often fail to transfer across model architectures. To address this, they propose LI-Boost,  which can help enhancement that increases the local invariance of perturbations via translated samples during gradient updates. Experiments are conducted mostly on ImageNet across a diverse suite of attacks, CNN and ViT models  and defenses, showing that LI-Boost enhances transferability over a broad type of attack families. However, the discussion of related literature is somehow limited and a deeper engagement with these works and how the proposed approach advances beyond them is lacked.

**Strengths:**

1. This work presents a clear method to help improve trasnferability and lists extensive empirical settings for reproducibility.

2. The proposed method is fairly general , which  can be incorporated into multiple categories of adversarial attacks.

**Weaknesses:**

1. The paper presents the main idea and method fairly clearly, but there are many languages and formatting issues, which detract from its professionalism. 1)  Citation formatting. For example, “ResNet-50 ((RN-50) He et al. (2016))” or “ViT( Dosovitskiy et al. (2021))”  or "ILA ( Huang et al. (2019))"-> "ILA (Huang et al., 2019)", which includes  unnecessary spaces and redundant  parentheses. This  occurs in multiple places.  2) Spelling and grammar. "diversifing"-> "diversifying", "an Large Language Model” -> "a Large Language Model”. 3) Spacing and punctuation.  There are several missing or inconsistent spaces, especially in section 2.  e.g., "transferability( Wang
et al. (2023c); Zhang et al. (2022a;b))".

2. The discussion of closest prior work is  not enough.  1. Dong et al. (2019)’s TIA directly targets  translation invariance, but only briefly mentioned rather than  analyzed critically.  2. some prior work ( [1], [2]) are highly relevant and should serve new baselines in augmentation-based transferability of adversarial attacks.

3. The paper introduces the concept of “translation invariance of adversarial perturbations”. As translation invariance is a well-established concept in CV area, here it seems repurposed mainly to describe the sensitivity of perturbations to pixel shifts. This term may be seen as over-formalized, since the underlying idea is intuitive and could be explained without introducing new jargon. I suggest clarifying whether this is simply an application of the standard concept of translation invariance, or if there is a genuinely novel definition that warrants a distinct term.

4. The experimental results is not through.  Other dataset should be tested (e.g., cifar10). See other details in the questions part.


[1] Yun, Zebin, et al. "The Ultimate Combo: Boosting Adversarial Example Transferability by Composing Data Augmentations." Proceedings of the 2024 Workshop on Artificial Intelligence and Security. 2024.

[2] Li, Qizhang, et al. "Towards evaluating transfer-based attacks systematically, practically, and fairly." Advances in Neural Information Processing Systems 36 (2023): 41707-41726.

**Questions:**

1. Can LI-Boost  consistently improve the SoTA combination of adversarial attack? e.g.,  Admix-TI-DIM+ LI-Boost > Admix-TI-DIM [3], UltComb_Gen [1]+LI-Boost > UltComb_Gen. If  the author can generate new results to show this, I am happy to reconsider my score.


2. If pixel shifting is applied to the target model when  test  adversarial examples, does it affect ASR?


[3] Wang, Xiaosen, et al. "Admix: Enhancing the transferability of adversarial attacks." Proceedings of the IEEE/CVF international conference on computer vision. 2021.

---

### Official Review · Reviewer_pAKH · 2025-10-31

**Soundness:** 2
**Presentation:** 4
**Contribution:** 1
**Rating:** 2
**Confidence:** 5

**Summary:**

This paper explores adversarial example transfer attacks. Starting from the translation invariance of CNNs, the authors introduce a Local Invariance Boosting approach (LI-Boost). While the motivation and experiments are not entirely accurate, the proposed method may have some effectiveness.

**Strengths:**

Exploring adversarial transferability is beneficial for the secure deployment of models.

**Weaknesses:**

1. The motivation behind this paper is seriously flawed. Lines 195-198 state that all DNNs possess translation invariance, which is incorrect. In fact, only CNNs possess translation invariance, a point supported by the cited references. This flawed assumption implies that the derived method is flawed.

2. We observe that this paper only selects CNN models for surrogate models, omitting ViT. Is this because ViT does not support translation invariance? This lack of translation invariance further limits the generalization ability of the proposed method, as most current models utilize transformer architectures rather than CNNs.

3. The proposed method introduces a significant computational cost and presents unfair comparisons. In this paper, $N=30$, meaning the gradient requires 30 additional forward and backward propagations, a substantial overhead. For example, MI-FGSM requires only one forward and backward propagation, while LI-Boost-MI requires 30, resulting in a significant performance improvement.

4. The improvement in adversarial transferability brought about by translation invariance lacks intuitive or theoretical explanation. We don't need a trick; we need to understand why doing so can improve adversarial transferability, thereby guiding network design. Quantitative indicators or qualitative visualization analysis should be provided here.

5. Translation invariance is not entirely novel. Random translation variations were introduced early on [1] to improve the generalization of adversarial examples.

6. There was a lack of motivation in choosing a sampling distribution. Why choose Logarithmic sampling, and what challenges did it address?

[1] Adversarial Patch, NeurIPS 2017 Workshop

**Questions:**

Please refer to Weaknesses.

**Details Of Ethics Concerns:**

This paper does not discuss the harmful risks that the proposed method poses to society and existing machine learning applications.

---

### Official Review · Reviewer_Qzqr · 2025-11-01

**Soundness:** 2
**Presentation:** 3
**Contribution:** 2
**Rating:** 4
**Confidence:** 5

**Summary:**

This paper investigates the relationship between local invariance and adversarial transferability.
The authors propose LI-Boost, a plug-in mechanism that enhances transferability by encouraging perturbations to remain effective under local translations. The method averages gradients from multiple translated perturbations during optimization. Experiments are conducted on ImageNet across CNNs, Vision Transformers (ViTs), and several defense models.

**Strengths:**

1.	The paper is well-organized and easy to follow, with a clear structure and logical presentation.

2.	The proposed method is simple, making it easy to understand and reproduce.

**Weaknesses:**

1.	The novelty of the proposed method is limited, as LI-Boost is similar to the well-known Translation-Invariant FGSM (TI-FGSM, Dong et al., CVPR 2019). Under reasonable assumptions, the mathematical formulations of the two methods are approximately equivalent.

2.	The claimed positive correlation between "local invariance" and transferability is supported only by empirical evidence, without theoretical reasoning to explain the underlying relationship.

3.	The experimental evaluation is incomplete: it lacks comparisons with other closely related baselines (such as TI-FGSM and similar attacks), omits baseline attack methods in the model ensemble experiments (e.g., Table 5 does not include those in Table 4), and in the appendix experiments that change source models, the baseline attacks are again missing.

4.	The paper does not provide any comparison of computational cost or runtime between LI-Boost and similar methods such as TI-FGSM, leaving the practical trade-off between performance and efficiency unexplored.

**Questions:**

1.	Could you clarify in detail how LI-Boost differs from TI-FGSM?

2.	Why are some baseline attacks (e.g., TI-FGSM) and base methods omitted in the experiments?

3.	Could you report runtime or computational cost comparisons between LI-Boost and existing methods such as TI-FGSM?

4.	Can you provide any theoretical reasoning or formal explanation for why "local invariance" should correlate positively with transferability?

---

### Note · Authors · 2025-11-12

I have read and agree with the venue's withdrawal policy on behalf of myself and my co-authors.